# Contrastive-and-Correlation Catalysts for Cross-Domain Long-Tail Class-incremental Learning

## Abstract

Long-tail class-incremental learning (LTCIL) requires models to sequentially learn new classes from long-tailed data distributions while mitigating catastrophic forgetting. However, existing methods predominantly assume that each incremental task originates from a single domain, neglecting the practical scenario where tasks involve heterogeneous domains. To address this limitation, we introduce **Cross-Domain LTCIL (CD-LTCIL)**, a challenging and underexplored setting where each task consists of data from multiple domains. We observe that conventional LTCIL methods exhibit significant performance degradation under cross-domain semantic shifts due to their limited domain generalization capability. To overcome these challenges, we propose **C2C** (Contrastive-and-Correlation Catalysts), a parameter-efficient framework that maintains a frozen pre-trained backbone and learns lightweight catalyst pathways. For the first task, C2C employs cosine anchoring combined with bi-level contrastive learning to establish domain-invariant class representations. For subsequent tasks, it preserves previously acquired knowledge through cross-correlation distillation between a frozen base catalyst and a learnable incremental catalyst. Extensive experiments on standard LTCIL benchmarks (CIFAR-100, ImageNet-R) and the proposed cross-domain Hybrid-DomainNet demonstrate that our approach achieves state-of-the-art performance across all evaluated scenarios, establishing a strong foundation for real-world long-tail continual learning under multi-domain conditions. The code will be made publicly available.

## 1 Introduction

The pursuit of human-like learning capabilities in artificial intelligence necessitates the development of systems that can continuously learn from non-stationary data streams. Class-incremental learning (CIL) (Van de Ven & Tolias, 2019) represents a fundamental paradigm in this endeavor, aiming to train models that sequentially acquire knowledge of new classes while maintaining performance on previously learned ones, thereby addressing the problem of catastrophic forgetting (McCloskey & Cohen, 1989). As shown in Figure 1, conventional CIL typically operates under the assumption that each incremental task contains balanced data drawn from a single domain. Recent studies (Liu et al., 2022; Kalla & Biswas, 2024; Gu et al., 2025; Qi et al., 2025) have extended this paradigm to long-tail CIL (LTCIL), where new classes follow long-tailed distributions characterized by a few head classes and many tail classes with limited samples. This inherent data imbalance introduces additional challenges, as models tend to exhibit bias toward newly learned classes, further exacerbating the forgetting of tail classes from previous tasks.

However, both conventional CIL and LTCIL remain constrained by an unrealistic assumption: *all incremental task share the same domain*. In practice, visual systems often acquire new knowledge from multiple heterogeneous domains. For instance, in real-world scenarios such as autonomous driving and medical diagnosis, models encounter data that is not only incrementally presented with long-tailed distributions but also originates from multiple heterogeneous domains (e.g., different camera viewpoints and medical imaging modalities). This critical gap motivates us to define and tackle a more intricate and realistic challenge: **Cross-Domain Long-Tail Class-Incremental**

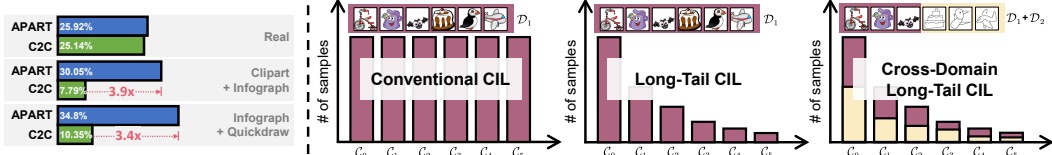

Figure 1: Performance comparison and learning settings. (Left) Performance gap between initial and final tasks for APART and C2C methods across three datasets. (Right) Illustration of the proposed CD-LTCIL setting. In contrast to traditional CIL and LTCIL, which assume single-domain incremental tasks, CD-LTCIL introduces a more realistic scenario where each task comprises images from multiple heterogeneous domains.

**Learning (CD-LTCIL)**, as illustrated in Figure 1. In this setting, each incremental task comprises samples sourced from multiple fixed heterogeneous domains, while also exhibiting a long-tailed distribution. As a result, CD-LTCIL confronts three primary challenges: ❶ *Catastrophic Forgetting:* The unavailability of previous task data causes the model to forget knowledge of classes learned from earlier domains; ❷ *Cross-Domain Shift:* Significant semantic discrepancies, induced by the mixture of heterogeneous domains, lead to pronounced performance degradation across domains; ❸ *Class Imbalance:* Driven by the long-tailed data distribution within tasks, the model becomes biased toward head classes, which significantly hinders its capability to recognize tail classes across the current and subsequent domains under distribution shift.

Existing methods address these issues in isolation. Replay-based (Aljundi et al., 2019; Liu et al., 2020; Bang et al., 2021) and regularization-based (Lopez-Paz & Ranzato, 2017; Tang et al., 2021; Wang et al., 2021) approaches mitigate catastrophic forgetting, but they typically assume balanced data and a single domain; Dynamic architecture expansion methods (Ostapenko et al., 2019; Yan et al., 2021; Wang et al., 2022a) allocate capacity to new tasks yet still presume data balance and domain homogeneity, thus failing to confront the dual challenges of domain shift and class imbalance; domain adaptation (DA) (Ganin & Lempitsky, 2015; Zheng et al., 2025) and domain generalization (DG) (Li et al., 2018; Pulakurthi et al., 2025) target distribution shift but are generally confined to static, non-incremental scenarios. Long-tailed methods focus primarily on class imbalance, overlooking cross-domain challenges and catastrophic forgetting. As shown in Figure 1, the performance gap of APART between initial and final tasks is significantly larger in the single-domain (Real) setting compared to cross-domain scenarios. In contrast, our method C2C suppressed this gap to merely 7.79% and 10.35% (reductions of 74.1% and 70.3%) on the DomainNet (Peng et al., 2019) subset combinations of Clipart+Infograph and Infograph+Quickdraw, respectively. Simply combining independent solutions is insufficient and may even introduce negative interference. For instance, replay buffers may store tail-class samples from old domains, but if used without regard to the feature space of new domains, such replay can amplify domain shift. Conversely, applying domain alignment without addressing within-task class imbalance may distort fragile decision boundaries for tail classes.

To this end, we make the following contributions:

- We propose a new setting, **Cross-Domain Long-Tail Class-Incremental Learning (CD-LTCIL)**, which removes the conventional single-domain assumption in LTCIL and better reflects real-world scenarios. To the best of our knowledge, this is the first work to explicitly address LTCIL across diverse domains.

- To tackle the CD-LTCIL problem, we propose C2C, an exemplar-free framework that integrates contrastive transfer, cross-correlation distillation, and logit-adjusted calibration in a parameter-efficient manner, enabling the model with the capability to simultaneously adapt to domain shifts and class increments while effectively addressing the challenge of class imbalance.

- Furthermore, We construct **Hybrid-DomainNet**, a new benchmark specifically designed for CD-LTCIL, providing a solid foundation for training and evaluating LTCIL models under diverse domains.

## 2 RELATED WORK

**Class-Incremental Learning.** As a core and challenging problem in continual learning (Van de Ven & Tolias, 2019), class-incremental learning (CIL) aims to enable models to learn new classes se-

quentially without forgetting previous knowledge. Mainstream CIL methods can be broadly fall into three main types: replay-based, regularization-based, and parameter isolation-based. Replay-based methods (Bang et al., 2021; Shibata et al., 2021; Shim et al., 2021) retain a small set of exemplars from past tasks or generate synthetic samples to rehearse old knowledge during new task learning. Regularization-based methods (Lopez-Paz & Ranzato, 2017; He & Jaeger, 2018; Tiwari et al., 2022) primarily aim to protect acquired knowledge by incorporating additional regularization constraints during training to restrict changes in model parameters. Parameter isolation-based methods (Aljundi et al., 2017; Mallya & Lazebnik, 2018) assign independent subsets of model parameters to each task, thereby avoiding interference between tasks. Recently, with the widespread adoption of pre-trained models (PTMs), PTM-based methods (Wang et al., 2022b;c; Smith et al., 2023) have gained increasing attention. These approaches typically design lightweight modules to efficiently adapt pre-trained models to different tasks in a parameter-effective manner.

**Long-Tail Learning.** Long-Tail learning is an important research direction in the fields of machine learning and computer vision, with its core objective being to address the prevalent long-tailed distribution problem in real-world data. Under such a distribution, class imbalance causes models to be biased toward head classes, leading to a sharp decline in recognition performance for tail classes. Existing methods can be broadly categorized into three paradims (Zhang et al., 2023): Class re-balancing, information augmentation, and module design. Class re-balancing methods (Chawla et al., 2002; Zhao et al., 2018; Zhang & Pfister, 2021) aim to design reasonable weighting strategies to mitigate the negative impact of class imbalance. Information augmentation methods (Cui et al., 2018; He et al., 2021a;b) seek to to enrich the representations of tail classes by introducing additional data or knowledge. In contrast, module design approaches (Huang et al., 2016; Ouyang et al., 2016) incorporate dedicated network components to handle the long-tail distribution effectively.

**Long-Tail Class-Incremental Learning.** Long-tail class-incremental learning (LTCIL) represents a more realistic and challenging extension of CIL, where models must handle both catastrophic forgetting and severe class imbalance within each incremental phase, as well as in the final evaluation over all classes. Current LTCIL methods can be divided into two types: two-stage replay-based methods and one-stage adapter-based methods. Among replay-based methods, LWS (Liu et al., 2022) proposed a learnable weight scaling layer to compensate for class imbalance. GVAlign (Kalla & Biswas, 2024) improved upon LWS by replaying generated pseudo-augmented samples to enhance representation robustness and align classifiers. In the adapter-based category, DAT Gu et al. (2025) employed a dynamic adapter caching mechanism to mitigate cross-task and cross-class disparities in LTCIL, alleviating catastrophic forgetting and imbalance. APART (Qi et al., 2025) designed separate adapter pools for head and tail classes, along with an adaptive routing strategy, to effectively address class imbalance. However, a common limitation of existing LTCIL methods is their reliance on the assumption that incremental tasks occur within a single, homogeneous domain. This restricts their applicability in dynamic real-world scenarios involving multiple domains. To bridge this gap, we propose the Cross-Domain LTCIL (CD-LTCIL) setting, which introduces domain shifts during the incremental process, thereby better aligning with practical conditions.

## 3 METHOD

In this section, we formalizes the problem of Cross-Domain Long-Tail Class-Incremental Learning (CD-LTCIL) and introduces our proposed framework, C2C (Contrastive-and-Correlation Catalysts). C2C is a parameter-efficient continual learner that operates by freezing a pre-trained backbone and augmenting it with lightweight "catalyst" branches. The framework operates in two key phases: during the first task, it transfers knowledge via contrastive alignment, which combines cosine anchoring to the frozen backbone with bi-level supervised contrastive learning. For subsequent tasks, it preserves prior knowledge through a correlation-based distillation mechanism between a frozen base catalyst and a learnable incremental catalyst, while expanding the classifier. The overall architecture is illustrated in Figure 2.

### 3.1 PROBLEM SETTING

Standard LTCIL is defined over a sequence of tasks denoted by $k = 1, 2, \ldots$. At task $k$, the model encounters a training set $\mathcal{S}_k = \{(\mathbf{u}_j, c_j)\}_{j=1}^{n_k}$, where $\mathbf{u}_j \in \mathbb{R}^{H \times W \times C}$ and labels $c_j \in \mathcal{C}_k$. The

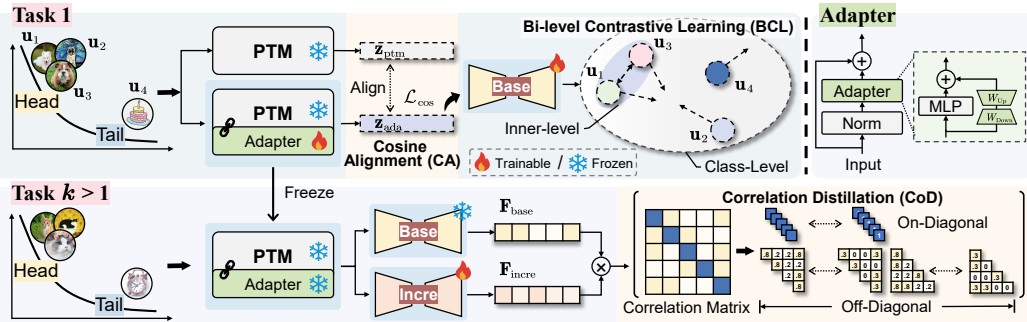

Figure 2: Overview of the proposed C2C framework. For Task 1, knowledge is transferred from the frozen pre-trained model to the learnable catalyst pathway to enhance generalization. For Task $k > 1$, the base catalyst is frozen, and the incremental catalyst is updated under distillation from the base to balance plasticity and stability.

label sets are disjoint across tasks, i.e., $\mathcal{C}_k \cap \mathcal{C}_{k'} = \varnothing$ for $k \neq k'$. The cumulative label space up to task $k$ is $\mathcal{C}_{1:k} = \bigcup_{m=1}^{k} \mathcal{C}_m$, and the class frequencies within each $\mathcal{S}_k$ follow a long-tailed distribution. The model at step $k$, $f_k : \mathbb{R}^{H \times W \times C} \to \mathbb{R}^{|\mathcal{C}_{1:k}|}$, is decoupled into a feature extractor $\psi : \mathbb{R}^{H \times W \times C} \to \mathbb{R}^d$ and an incrementally expanding classifier $h_k : \mathbb{R}^d \to \mathbb{R}^{|\mathcal{C}_{1:k}|}$, such that $f_k = h_k \circ \psi$. Under the exemplar-free condition, training at step $k$ utilizes only $\mathcal{S}_k$. During inference, the model receives no task identity and is evaluated on the balanced test union $\mathcal{T}_{1:k} = \bigcup_{m=1}^{k} \mathcal{T}_m$ via the aggregate risk $\sum_{(\mathbf{u},c) \in \mathcal{T}_{1:k}} \ell(f_k(\mathbf{u}), c)$.

A key limitation of the standard Long-Tail Class-Incremental Learning (LTCIL) formulation is its assumption that data originates from a single domain throughout the learning process. This assumption often contradicts real-world situations where streaming data comes from multiple, heterogeneous domains. To bridge this gap, we introduce the Cross-Domain LTCIL (CD-LTCIL) setting. In CD-LTCIL, at each training step $k$, data are drawn from a fixed collection of heterogeneous domains $\Omega$, constituting a dataset $\mathcal{S}_k^{\Omega} = \{(\mathbf{u}_j, c_j)\}_{j=1}^{n_k}$. Although each instance $\mathbf{u}_j$ belongs to a domain $r_j \in \Omega$, *domain identifiers are not utilized during training*. Notably, the domain set is consistent over all tasks (that is, $|\Omega_k| = |\Omega|$ and $\Omega_k = \Omega \ \forall k$), while the class sets are disjoint across tasks ($\mathcal{C}_k \cap \mathcal{C}_{k'} = \varnothing$ for $k \neq k'$). The goal is to learn a model $f_k^{\Omega} : \mathbb{R}^{H \times W \times C} \to \mathbb{R}^{|\mathcal{C}_{1:k}|}$, which consists of a domain-robust feature extractor $\psi$ and an incrementally expanded classifier $h_k$. This model should preserve knowledge from previous tasks and maintain generalization across domains, even under long-tailed class distributions. For evaluation, the model is tested on a balanced cross-domain test set $\mathcal{T}_{1:k}^{\Omega} = \bigcup_{m=1}^{k} \bigcup_{r \in \Omega} \mathcal{T}_m^{(r)}$, with performance measured by $\sum_{(\mathbf{u},c,r) \in \mathcal{T}_{1:k}^{\Omega}} \ell(f_k^{\Omega}(\mathbf{u}), c)$. It is important to emphasize that the learning process has no access to previous training data or domain identifiers, both during training and inference.

## 3.2 C2C ARCHITECTURE

To preserve task-agnostic knowledge, we keep the pre-trained backbone $\phi(\cdot)$ frozen. We then attach lightweight residual *catalyst* blocks into each layer, which, along with the task-specific classifier, remain trainable. Each catalyst is implemented as a post-normalization bottleneck module with down- and up-projections on the residual path. Formally, let $g(\cdot; \theta)$ denote a catalyst parameterized by $\theta$. The adapted feature representation is computed as:

$$\mathbf{z} = \text{Norm}\big(\phi(\mathbf{u})\big) + g\big(\text{Norm}(\phi(\mathbf{u})); \theta\big) \in \mathbb{R}^d. \tag{1}$$

The feature $\mathbf{z}$ is subsequently passed through a projection head, which functions as the *base* projector in Task 1 and is adapted to become the *incremental* projector in subsequent tasks.

This design presents key advantages for the CD-LTCIL setting: (i) Freezing most parameters and avoiding exemplar retention inherently mitigates catastrophic forgetting. (ii) The compact task-specific catalyst $g$ enables adaptation to heterogeneous domains with minimal perturbation to the stable features from $\phi$. (iii) The frozen backbone provides a stable feature foundation that is par-

ticularly beneficial for tail classes, while the localized parameter updates prevent overfitting to head classes under long-tailed distributions.

The classifier $h_k$ is expanded from $|\mathcal{C}_{1:k-1}|$ to $|\mathcal{C}_{1:k}|$ at each step $k$. Only the parameters $\theta$ of the catalyst and the newly added columns of the classifier are optimized during each incremental step.

## 3.3 CONTRASTIVE TRANSFER FROM THE PRE-TRAINED MODEL

The objective during the first task ($k = 1$) is to learn a domain-robust representation using only the current data, while preventing significant deviation from the beneficial geometric structure of the pre-trained model.

**Cosine Alignment (CA) to PTM.** We align features from the catalyst pathway with features extracted directly from the frozen backbone (with the catalyst suppressed). Let $\mathbf{z}_{\text{ada}}$ and $\mathbf{z}_{\text{ptm}}$ denote the adapted and direct pre-trained features, respectively. After standardization and $\ell_2$-normalization, we minimize the following objective:

$$\mathcal{L}_{\cos} = \frac{1}{B} \sum_{i=1}^{B} \big(1 - \langle \hat{\mathbf{z}}_{\text{ptm}}^{(i)}, \hat{\mathbf{z}}_{\text{ada}}^{(i)} \rangle\big). \tag{2}$$

This loss is designed to retain the generally domain-invariant knowledge of the pre-trained model, while allowing the catalyst to learn task-specific residual variations.

**Bi-level Contrastive Learning (BCL).** Given the absence of domain labels and exemplars throughout the entire learning process, we employ within-batch supervision to shape robust inter- and intra-class structures. For a mini-batch $\{\mathbf{z}^{(i)}, y^{(i)}\}_{i=1}^{B}$, we define the similarity as $S_{ij} = \langle \hat{\mathbf{z}}^{(i)}, \hat{\mathbf{z}}^{(j)} \rangle / \tau_s$. First, to encourage compact class clusters, we treat all same-class pairs as positives:

$$\mathcal{L}_{\text{class}} = -\frac{1}{B} \sum_{i=1}^{B} \log \frac{\sum_{j \neq i,\, y^{(j)} = y^{(i)}} \exp(S_{ij})}{\sum_{j \neq i} \exp(S_{ij})}. \tag{3}$$

Second, to mitigate potential sub-domain fragmentation within a class, we apply an inner-level contrastive loss using the top-$K$ nearest neighbors among same-class samples:

$$\mathcal{L}_{\text{inner}} = -\frac{1}{B} \sum_{i=1}^{B} \log \frac{\sum_{j \in \mathcal{N}_K(i)} \exp(S_{ij})}{\sum_{j \neq i} \exp(S_{ij})}. \tag{4}$$

These two losses are particularly suitable for CD-LTCIL as they promote class separation that is robust to domain shifts and encourage compact intra-class representations without relying on explicit domain labels.

## 3.4 CORRELATION DISTILLATION

For subsequent tasks ($k > 1$), we freeze the *base* catalyst (learned at $k = 1$) and introduce a new *incremental* catalyst. To preserve knowledge without exemplars, we leverage higher-order statistical relationships. Let $\mathbf{F}_{\text{base}}, \mathbf{F}_{\text{incre}} \in \mathbb{R}^{B \times d}$ be batch features from the frozen base and learnable incremental catalysts, respectively. We apply per-dimension whitening and compute the correlation matrix as follows:

$$\tilde{\mathbf{F}} = \frac{\mathbf{F} - \mathbf{1}\mu^{\top}}{\sigma^{\top}}, \qquad \mathbf{C} = \frac{1}{B} \tilde{\mathbf{F}}_{\text{incre}}^{\top} \tilde{\mathbf{F}}_{\text{base}} \in \mathbb{R}^{d \times d}. \tag{5}$$

The distillation loss is then defined as:

$$\mathcal{L}_{\text{corr}} = \sum_{i=1}^{d} (1 - C_{ii})^2 + \sum_{i \neq j} w_{ij}\, C_{ij}^2. \tag{6}$$

The first term encourages one-to-one alignment between corresponding feature dimensions to maintain stability. The second term penalizes off-diagonal correlations to prevent representational collapse and preserve plasticity for learning new concepts.

**Weighted Off-diagonal Penalty.** We employ an adaptive weighting scheme for the off-diagonal penalties:

$$w_{ij} = \begin{cases} \lambda_{\text{topk}}, & (i \neq j) \wedge (i,j) \in \mathcal{K}, \\ \lambda_{\text{rest}}, & (i \neq j) \wedge (i,j) \notin \mathcal{K}, \end{cases} \tag{7}$$

where $\mathcal{K}$ contains the indices of the $K$ largest absolute values in the off-diagonal part of $\mathbf{C}$. This strategy applies a smaller penalty ($\lambda_{\text{topk}}$) to the strongest off-diagonal correlations, which may encode important shared semantics, while applying a larger penalty ($\lambda_{\text{rest}}$) to the multitude of weaker, potentially noisy correlations. This helps avoid over-orthogonalization of the feature space.

### 3.5 Classification Objective and Training Schedule

Supervision at each step $k$ is restricted to the classes present in $\mathcal{S}_k$. Given the full logit vector $\mathbf{o} \in \mathbb{R}^{|\mathcal{C}_{1:k}|}$, we slice the portion corresponding to the current task, $\mathbf{o}^{\text{cur}} = \mathbf{o}_{[|\mathcal{C}_{1:k-1}|:|\mathcal{C}_{1:k}|]}$, and minimize the cross-entropy loss:

$$\mathcal{L}_{\text{ce}} = \frac{1}{B} \sum_{i=1}^{B} \text{CE}\big(\mathbf{o}^{\text{cur}(i)}, \; y^{(i)} - |\mathcal{C}_{1:k-1}|\big). \tag{8}$$

This approach ensures that gradients are only computed with respect to the available current-task labels. The overall training schedule is : (i) Task 1 Optimize $\mathcal{L}_{\text{ce}} + \lambda_{\cos}\mathcal{L}_{\cos} + \alpha_{\text{c}}\mathcal{L}_{\text{class}} + \alpha_{\text{i}}\mathcal{L}_{\text{inner}}$. The trained catalyst is saved as the base catalyst. (ii) Tasks $k > 1$ Freeze the base catalyst and optimize $\mathcal{L}_{\text{ce}} + \beta\,\mathcal{L}_{\text{corr}}$ for the incremental catalyst.

### 3.6 Long-tail Calibration

To address the class imbalance within each $\mathcal{S}_k$ without exemplars, we employ a logit-adjusted cross-entropy loss applied only to the current task's logits:

$$\mathcal{L}_{\text{lace}} = \frac{1}{B} \sum_{i=1}^{B} \text{CE}\big(\mathbf{o}^{\text{cur}(i)} + \tau \log \nu, \; y^{(i)} - |\mathcal{C}_{1:k-1}|\big), \tag{9}$$

where $\nu$ is the vector of empirical class frequencies in $\mathcal{S}_k$. This adjustment introduces a principled prior based on the label distribution. The same adjustment can be applied during inference by adding $\tau \log \nu$ to the current-task logits.

## 4 Experiments

### 4.1 Experimental Setup

**Datasets and Evaluation.** Following the standard LTCIL setting and prior works Liu et al. (2022); Qi et al. (2025), we evaluate our method on **CIFAR-100** (Krizhevsky, 2009), **ImageNet-R** (Hendrycks et al., 2021), and **DomainNet** (Peng et al., 2019). CIFAR-100 comprises 100 classes with 60,000 images, while ImageNet-R contains 200 classes with 30,000 images. To simulate a long-tailed distribution, the imbalance ratio is controlled by $\rho$, defined as the ratio of the sample size of the least frequent class to that of the most frequent class. For CIFAR-100, we set $\rho = 0.01$, with the largest class containing 500 samples. ImageNet-R inherently exhibits a long-tailed distribution with $\rho = 0.11$ and a maximum of 349 samples per class. Although its distribution does not follow a standard exponentially decaying pattern, we retain the original data distribution.

For the CD-LTCIL setting, we introduce **Hybrid-DomainNet**, a dataset derived from DomainNet containing 345 classes from six domains (Clipart, Infograph, Real, Quickdraw, Sketch, Painting). It comprises four partitions: Clipart+Infograph, Clipart+Quickdraw, Infograph+Quickdraw, and a combination of all three, derived from three discriminative DomainNet subsets. The base and incremental tasks follow a fixed multi-domain long-tailed distribution, similar to LTCIL. Construction details are in Appendix D.

Dataset partitioning follows Liu et al. (2022); Qi et al. (2025): the initial task contains half of the classes, and the remaining classes are divided into 5 or 10 incremental tasks. This paper focuses on

ordered long-tailed scenarios. Following Rebuffi et al. (2017), we record the accuracy after each task and use the average accuracy over all tasks as the evaluation metric. Our implementation is based on PyTorch (Paszke et al., 2019). All experiments were conducted on a single NVIDIA GeForce RTX 4090 GPU, and the final results are averaged over three random splits.

**Implementation Details.** Consistent with Wang et al. (2022c); Qi et al. (2025), we employ the ViT-B/16 backbone (Dosovitskiy et al., 2020) pre-trained on ImageNet21K (Russakovsky et al., 2015). The embedding dimension is $d=768$. To ensure a fair comparison, the same backbone architecture is used for all methods. We use AdamW optimizer (weight decay $5\times10^{-4}$) with a batch size of 48 and train for 7 epochs per task. The initial learning rate is $3\times10^{-3}$ and follows a cosine annealing schedule with $\eta_{\min}=10^{-8}$. For prompt-based baselines, the prompt-pool size is set to 10; for replay-based methods LUCIR (Hou et al., 2019) and LWS (Liu et al., 2022), we use 10 exemplars per class. APART (Qi et al., 2025) is reproduced with a pool size of 5 and a projection dimension of 64, following its original configuration. Further C2C-specific settings are provided in Appendix A.

### 4.2 COMPARISONS WITH STATE-OF-THE-ARTS

**Compared Methods.** We compare our method against state-of-the-art LTCIL approaches under the CD-LTCIL setting. All methods have been reimplemented to ensure a fair comparison. The compared methods include: conventional CIL methods LwF (Li & Hoiem, 2017) and SimpleCIL (Zhou et al., 2025); recent prompt-based tuning approaches L2P (Wang et al., 2022c), DualPrompt (Wang et al., 2022b), and CODA-Prompt (Smith et al., 2023); and standard LTCIL methods LUCIR (Hou et al., 2019), LUCIR+LWS (Liu et al., 2022), and APART (Qi et al., 2025). Experimental results for cross-domain tasks are presented in Table 1, while single-domain results are shown in Table 2. Additional results and analyses are provided in Appendix B.

Table 1: Comparison with SOTA methods under the **CD-LTCIL** setting. "Info." denotes "Infograph"; "Quick." denotes "Quickdraw". Best results are **bolded**; second-best are underlined.

| Method | Replay | Clipart+Info. | | Clipart+Quick. | | Info.+Quick. | | Clipart+Info.+Quick. | |
|---|---|---|---|---|---|---|---|---|---|
| | | 5 Tasks | 10 Tasks | 5 Tasks | 10 Tasks | 5 Tasks | 10 Tasks | 5 Tasks | 10 Tasks |
| LwF | w/o | 42.18 | 38.92 | 45.63 | 41.27 | 32.15 | 28.74 | 36.84 | 32.59 |
| SimpleCIL | w/o | 44.37 | 40.85 | 47.92 | 43.56 | 34.28 | 30.63 | 39.12 | 35.04 |
| LUCIR | w/ | 48.26 | 45.73 | 50.47 | 48.35 | 38.42 | 35.91 | 43.67 | 40.28 |
| LUCIR+LWS | w/ | 49.14 | 47.86 | 51.29 | 50.12 | 40.75 | 38.24 | 45.93 | 42.67 |
| L2P | w/o | 47.35 | 46.82 | 50.18 | 50.47 | 36.94 | 36.61 | 42.15 | 40.28 |
| DualPrompt | w/o | 46.82 | 46.17 | 49.63 | 49.28 | 35.83 | 34.45 | 41.04 | 40.13 |
| CODA-Prompt | w/o | 48.07 | 47.64 | 50.74 | 49.83 | 39.16 | 38.92 | 44.38 | 41.71 |
| APART | w/o | 50.93 | 52.57 | 51.95 | 58.04 | 40.11 | 45.67 | 47.36 | 48.82 |
| **C2C (Ours)** | w/o | **60.76** | **60.41** | **58.26** | **64.88** | **46.27** | **50.39** | **51.00** | **52.10** |

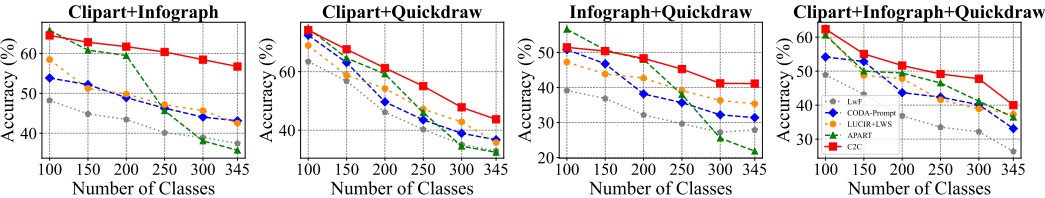

Figure 3: Comparison of the trend of classification accuracy between C2C and SOTA methods across all tasks under the **CD-LTCIL** setting.

**Comparisons under the CD-LTCIL Setting.** Table 1 presents a comprehensive comparison on Hybrid-DomainNet under the CD-LTCIL setting, reporting the average accuracy over 5-task and 10-task schedules. C2C consistently outperforms all prior works across all domain mixtures. Specifically, compared to the strongest prior baseline APART (also non-replay), C2C achieves improvements of **+9.83%/+7.84%** on Clipart+Infograph (5/10 tasks), **+6.31%/+6.84%** on Clipart+Quickdraw, **+6.16%/+4.72%** on Infograph+Quickdraw, and **+3.64%/+3.28%** on the three-domain mixture (Clipart+Infograph+Quickdraw). Averaged across the four mixtures, the gains are **+6.49%** (5 tasks) and **+5.67%** (10 tasks), yielding a macro average improvement of **+6.08%** over

APART. These results demonstrate that the proposed contrastive transfer at Task 1 and correlation distillation for subsequent tasks provide consistent and significant benefits for cross-domain incremental learning under long-tailed class distributions.

Figure 3 further illustrates the superior performance of C2C by showing the accuracy trends across all tasks. It can be observed that APART exhibits a noticeable declining trend, indicating its susceptibility to catastrophic forgetting. In contrast, LUCIR+LWS, which utilizes an exemplar replay mechanism, along with other PEFT-based techniques (e.g., L2P, DualPrompt, CODA-Prompt), demonstrate a more moderate performance decline. However, a significant performance gap is evident in the initial tasks, underscoring the effectiveness of the proposed CA and CoD strategies.

**Comparisons under the Standard LTCIL Setting.** To further validate the generalization capability of C2C, We evaluate it under the standard LTCIL setting on four benchmarks, comprising two full datasets (CIFAR-100 and ImageNet-R) and two domains (Real and Clipart) from DomainNet. As shown in Table 2, C2C consistently outperforms all existing SOTA methods. These results robustly demonstrate that C2C not only excels in challenging cross-domain scenarios but also achieves sustained performance improvements in standard LTCIL tasks.

Table 2: Comparison with SOTA methods under the **LTCIL** setting. Best results are **bolded**; second-best are underlined.

| Method | Replay | CIFAR-100 | | ImageNet-R | | Real | | Clipart | |
|---|---|---|---|---|---|---|---|---|---|
| | | 5 Tasks | 10 Tasks | 5 Tasks | 10 Tasks | 5 Tasks | 10 Tasks | 5 Tasks | 10 Tasks |
| LwF | w/o | 77.14 | 70.12 | 78.20 | 75.53 | 68.32 | 62.15 | 63.27 | 58.41 |
| SimpleCIL | w/o | 72.36 | 72.22 | 57.32 | 57.39 | 71.45 | 65.83 | 66.52 | 61.27 |
| LUCIR | w/ | 83.52 | 79.94 | 80.32 | 77.51 | 76.28 | 72.45 | 71.83 | 68.27 |
| LUCIR+LWS | w/ | 83.24 | 80.59 | 80.46 | 78.03 | 78.92 | 75.63 | 74.26 | 71.45 |
| L2P | w/o | 78.37 | 74.68 | 74.95 | 72.75 | 75.21 | 74.64 | 70.38 | 69.85 |
| DualPrompt | w/o | 77.88 | 75.80 | 72.85 | 71.24 | 74.85 | 73.02 | 68.91 | 66.17 |
| CODA-Prompt | w/o | 80.44 | 75.36 | 79.94 | 78.33 | 77.58 | 74.12 | 72.92 | 71.36 |
| APART | w/o | 87.16 | 84.21 | 81.41 | 78.92 | 81.45 | 85.33 | 76.50 | 79.73 |
| **C2C (Ours)** | w/o | **88.75** | **86.71** | **84.47** | **81.84** | **83.24** | **88.09** | **78.42** | **80.91** |

## 4.3 Ablation Study

**Effectiveness of Key Components.** We conduct a comprehensive ablation study to evaluate the contribution of each proposed component. As summarized in Table 3, incorporating cosine alignment (CA) into the baseline yields consistent improvements across both single-domain (Real, IN-R) and cross-domain (Clipart+Infograph, Infograph+Quickdraw) benchmarks. This demonstrates that anchoring the learnable catalyst to the frozen backbone geometry promotes domain-invariant feature learning. The addition of bi-level contrastive learning (BCL) further enhances performance by jointly enforcing inter-domain class compactness and mitigating intra-class feature dispersion. When correlation distillation (CoD) is introduced to handle incremental tasks, additional gains are observed, indicating its effectiveness in preserving critical inter-feature relationships between base and incremental catalysts. The complete C2C model, integrating all three components, achieves the best performance across all experimental settings.

Table 3: Ablation study of proposed components. IN-R denotes "ImageNet-R".

| Method | CA | BCL | CoD | Real | IN-R | Clipart+Info. | Info.+Quick. |
|---|---|---|---|---|---|---|---|
| Baseline | - | - | - | 76.22 | 78.58 | 51.72 | 40.34 |
| Propose | ✓ | | | 79.14 | 80.58 | 53.93 | 43.05 |
| | ✓ | | ✓ | 82.47 | 83.57 | 58.31 | 45.16 |
| | ✓ | ✓ | | 82.20 | 82.56 | 59.42 | 45.88 |
| | ✓ | ✓ | ✓ | **83.24** | **84.47** | **60.76** | **46.27** |

**Analysis of Bi-level Contrastive Learning.** Table 4 presents an ablation study on the components of BCL. The class-level contrastive term aggregates samples sharing the same label across different domains, thereby enhancing the consistency of class prototypes. The instance-level term focuses on the most relevant within-class neighbors (top-$K$), effectively suppressing sub-domain variations and yielding more compact decision boundaries. The combination of both levels demonstrates superior cross-domain generalization capability.

**Analysis of Correlation Distillation.** We further analyze the components of CoD in Table 5. The on-diagonal term encourages direct correspondence between incremental and base feature dimensions, facilitating the retention of previously acquired knowledge. The off-diagonal term employs an adaptive weighting scheme that imposes smaller penalties on the strongest cross-dimension correlations while applying larger penalties to weaker ones, thereby reducing interference while preserving shared structural information. The integration of both terms achieves the optimal balance between stability and plasticity during incremental learning.

Table 4: Ablation study of BCL components.

| Method | Clipart+Info. | Info.+Quick. |
|---|---|---|
| Baseline ( w/ CA) | 53.93 | 43.05 |
| + Class-level | 57.85 | 44.07 |
| + Inner-level | **59.42** | **45.88** |

Table 5: Ablation study of CoD components.

| Method | Clipart+Info. | Info.+Quick. |
|---|---|---|
| Baseline ( w/ CA) | 53.93 | 43.05 |
| + On-diagonal | 56.52 | 44.60 |
| + Off-diagonal | **58.31** | **45.16** |

**Analysis of Long-tail Calibration.** We evaluate the effectiveness of logit-adjusted calibration (LTC) by comparing models trained with and without the prior-based logit adjustment on current-task data. On Clipart+Infograph, accuracy improves from 59.30% to 60.76% with LTC; similarly, on Infograph+Quickdraw, it increases from 45.82% to 46.27%. These consistent improvements demonstrate that incorporating empirical class-frequency priors effectively mitigates head-tail bias.

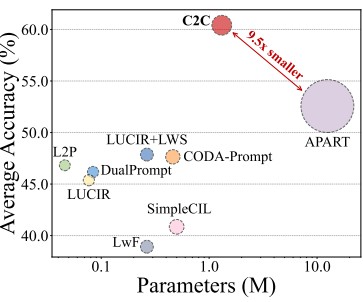
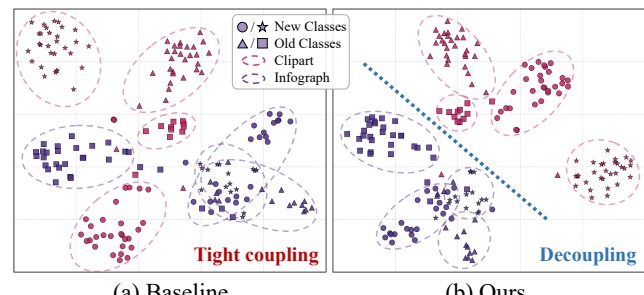

(a) Baseline    (b) Ours

Figure 4: Accuracy versus parameters for SOTA methods.

Figure 5: t-SNE visualization of feature embeddings. Colors represent different domains.

**Visualization Analysis.** Figure 4 compares the parameter efficiency of different methods. Unlike replay-based approaches that store exemplars, our method incurs storage overhead only from parameter-efficient tuning. Despite introducing additional parameters, this overhead remains minimal compared to the full pre-trained model. C2C achieves superior performance while maintaining a parameter budget comparable to existing prompt-based methods.

To validate the generalization capability of BCL across categories and domains, we visualize embeddings of both seen and unseen classes after adaptation on Clipart+Infograph using t-SNE (Maaten & Hinton, 2008). As shown in Figure 5a, BCL produces more separable feature distributions compared to the baseline, confirming its effectiveness at the class level. Furthermore, BCL clearly separates different domains (Figure 5b), demonstrating its strong domain generalization capability through the instance-level contrastive mechanism.

## 5 CONCLUSION

In this paper, we introduced the problem of Cross-Domain Long-Tail Class-Incremental Learning (CD-LTCIL) and proposed **C2C**, a parameter-efficient framework that learns lightweight catalyst pathways while keeping the pre-trained backbone frozen. Our approach employs cosine anchoring and bi-level supervised contrastive learning during the initial task, followed by cross-correlation distillation for subsequent incremental tasks, with logit-adjusted calibration to address class imbalance. Extensive experiments under both CD-LTCIL and standard LTCIL settings validate the effectiveness and generalization capability of the proposed method. We believe that the new problem formulation and solution presented in this work will inspire future research and contribute to advancing the field of cross-domain incremental learning.

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

# Appendix

## CONTENTS

## A IMPLEMENTATION DETAILS

This section provides comprehensive implementation details for our proposed C2C framework. Unless otherwise specified, all hyperparameters are set to their default values as detailed in Tables 6–8. We report average accuracy (%) on Hybrid-DomainNet, specifically on the Clipart+Infograph and Infograph+Quickdraw domain combinations (5 tasks in total).

### A.1 CONTRASTIVE TRANSFER LEARNING (CA + BCL) FOR TASK 1

For the first task, we jointly optimize the classification objective with two complementary components: cosine alignment (CA) to preserve the pre-trained knowledge, and bi-level supervised contrastive learning (BCL) to enhance feature discriminability. The cosine alignment loss is defined in Equation 2. The BCL objective comprises two terms: a class-level contrastive term (Equation 3) that promotes compactness within classes across domains, and an inner-level term (Equation 4) that identifies the top-$K$ nearest neighbors within each class to reduce sub-domain fragmentation.

### A.2 CORRELATION DISTILLATION (CoD) FOR INCREMENTAL TASKS

When learning new classes in subsequent tasks, we freeze the base catalyst learned during Task 1 and train an incremental catalyst by minimizing the correlation distillation loss defined in Equation 6. This loss incorporates adaptive off-diagonal weighting as specified in Equation 7. Prior to correlation computation, features undergo per-dimension standardization.

Table 6: Hyperparameter configurations for Task 1.

| Component | Symbol | Default | Range | Notes |
|---|---|---|---|---|
| Cosine alignment weight | $\lambda_{\cos}$ | 0.1 | [0.01, 0.30] | Controls adaptation to PTM; larger $\rightarrow$ more stability |
| Class-level contrast weight | $\alpha_c$ | 0.1 | [0.05, 0.30] | Promotes cross-domain class compactness |
| Inner-level contrast weight | $\alpha_i$ | 0.1 | [0.05, 0.30] | Mitigates sub-domain fragmentation |
| Similarity temperature | $\tau_s$ | 0.07 | [0.05, 0.20] | In $S_{ij} = \langle \hat{z}^{(i)}, \hat{z}^{(j)} \rangle / \tau_s$ |
| Inner-level neighbors | $K$ | 2 | {1, 2, 3, 5} | Top-$K$ nearest within-class neighbors |
| Batch size | $B$ | 48 | {32, 48, 64, 96} | Impacts contrastive statistics |

Table 7: Hyperparameter configurations for incremental task learning with CoD.

| Component | Symbol | Default | Range | Notes |
|---|---|---|---|---|
| Correlation loss weight | $\beta$ | 0.3 | [0.10, 0.70] | Stability–plasticity trade-off |
| Off-diagonal base penalty | $\lambda_{\text{rest}}$ | $5 \times 10^{-3}$ | $[1 \times 10^{-4}, 1 \times 10^{-2}]$ | Broad suppression of cross-dim corr. |
| Top-$K$ strong off-diagonals | $|\mathcal{K}|$ | 20 | {10, 20, 50, 100} | Indices of largest $|C_{ij}|$ |
| Top-$K$ reduced penalty | $\lambda_{\text{topk}}/\lambda_{\text{rest}}$ | 0.2 | [0.1, 0.5] | Down-weights few strong correlations |
| Feature standardization | — | per-batch | {per-batch, EMA(0.9)} | Mean/var for whitening |

## A.3 CLASSIFICATION OBJECTIVE AND LONG-TAIL CALIBRATION

We employ an expanding classifier formulation with the likelihood function defined in Equation 8. To address within-task class imbalance, we apply logit-adjusted cross-entropy on the current task data as specified in Equation 9. The same adjustment is applied during inference.

Table 8: Hyperparameters for classification and long-tail calibration.

| Component | Symbol | Default | Range | Notes |
|---|---|---|---|---|
| Cross-entropy on slice | — | — | — | Expanding classifier, see Equation 8 |
| Class-frequency prior | $\nu$ | empirical | empirical | Estimated on $\mathcal{S}k$ |
| Logit-adjust temperature | $\tau$ | 1.0 | [0.5, 2.0] | Strength of prior shift $\nu$ |

## A.4 EXPERIMENTAL SETUP AND REPRODUCIBILITY

All experiments were conducted on a single NVIDIA GeForce RTX 4090 GPU. We report mean performance across three independent runs with different random seeds for data partitioning. For methods requiring exemplar storage (e.g., LUCIR, LUCIR+LWS), we follow the evaluation protocol established in APART (Qi et al., 2025), ensuring fair comparison under equivalent memory constraints while using the same ViT-B/16 backbone architecture.

## B ADDITIONAL ABLATION STUDIES

### B.1 GROUP-WISE PERFORMANCE ANALYSIS

Following Qi et al. (2025), we categorize classes into three groups based on their sample sizes and evaluate performance separately. Table 9 presents results on CIFAR100 for many-shot ($\geq 100$ samples), medium-shot (20–100 samples), and few-shot ($\leq 20$ samples) classes.

**Notably**, even without incorporating the long-tail calibration loss $\mathcal{L}_{\text{lace}}$, our method surpasses existing state-of-the-art approaches. This demonstrates that C2C achieves comprehensive improvements across all class groups, with particularly significant gains in the challenging few-shot setting.

## C HYPERPARAMETER SENSITIVITY ANALYSIS

We also present a comprehensive sensitivity analysis of key hyperparameters. Unless specified otherwise, all experiments use the default values from Tables 6–8 and report average accuracy (%) on Hybrid-DomainNet with Clipart+Infograph and Infograph+Quickdraw domain combinations (5 tasks total).

Table 9: Group-wise classification accuracy (%) on CIFAR100. Best results are **bolded**.

| Method | Average | Many | Median | Few |
|---|---|---|---|---|
| LwF | 77.14 | 85.21 | 76.35 | 42.18 |
| SimpleCIL | 72.36 | 82.95 | 71.84 | 38.26 |
| LUCIR | 83.52 | 90.63 | 82.71 | 49.87 |
| LUCIR+LWS | 83.24 | 90.42 | 82.50 | 49.35 |
| L2P | 78.37 | 86.92 | 77.65 | 44.21 |
| DualPrompt | 77.88 | 86.54 | 77.20 | 43.79 |
| CODA-Prompt | 80.44 | 88.73 | 79.85 | 47.62 |
| APART | 87.16 | 93.34 | 87.00 | 53.90 |
| C2C w/o $\mathcal{L}_{\text{lace}}$ | 87.88 | 93.65 | 87.94 | 55.70 |
| **C2C (Ours)** | **88.75** | **93.72** | **88.05** | **57.83** |

As shown in Table 10, the cosine alignment weight achieves optimal performance at $\lambda_{\cos}$=0.10. Both smaller (0.05) and larger values (0.20, 0.30) result in degraded performance, indicating the importance of balancing pre-trained model preservation with feature adaptation. Table 11 demonstrates the effect of contrast temperature, where $\tau_s$=0.07 yields the best results. Lower temperatures (0.05) lead to over-concentrated similarity distributions, while higher values (0.10–0.20) cause under-concentration, both slightly degrading performance. The grid search over BCL loss weights in Table 12 reveals that the balanced configuration $\alpha_c$=$\alpha_i$=0.10 performs best, suggesting that class-level compactness and inner-level neighbor consistency require comparable weighting for optimal feature learning. Regarding neighbor selection, Table 13 indicates that $K$=2 provides the strongest performance. While $K$=1 offers insufficient robustness, larger values (3–5) begin to blur fine-grained intra-class distinctions.

Table 10: Sensitivity analysis of $\lambda_{\cos}$ (CA weight).

| $\lambda_{\cos}$ | Clipart+Info. | Info.+Quick. |
|---|---|---|
| 0.05 | 53.41 | 42.78 |
| **0.10** | **53.93** | **43.05** |
| 0.20 | 53.86 | 42.98 |
| 0.30 | 53.07 | 42.52 |

Table 11: Sensitivity analysis of contrast temperature $\tau_s$ (BCL).

| $\tau_s$ | 0.05 | **0.07** | 0.10 | 0.15 | 0.20 |
|---|---|---|---|---|---|
| Clipart+Info. | 59.10 | **59.42** | 59.01 | 58.47 | 57.82 |
| Info.+Quick. | 45.58 | **45.88** | 45.67 | 45.11 | 44.35 |

Table 12: Sensitivity analysis of BCL loss weights $\alpha_c$ and $\alpha_i$ ($K$=2).

| $\alpha_c \backslash \alpha_i$ | Clipart+Info. | | | Info.+Quick. | | |
|---|---|---|---|---|---|---|
| | 0.05 | **0.10** | 0.20 | 0.05 | **0.10** | 0.20 |
| 0.05 | 58.77 | 59.12 | 58.81 | 45.05 | 45.39 | 45.11 |
| **0.10** | 59.08 | **59.42** | 59.06 | 45.41 | **45.88** | 45.52 |
| 0.20 | 58.70 | 59.05 | 58.68 | 45.03 | 45.48 | 45.09 |

Table 13: Sensitivity analysis of inner-level neighbors $K$ (BCL).

| $K$ | Clipart+Info. | Info.+Quick. |
|---|---|---|
| 1 | 59.11 | 45.62 |
| **2** | **59.42** | **45.88** |
| 3 | 59.27 | 45.73 |
| 5 | 58.88 | 45.30 |

The stability–plasticity trade-off in correlation distillation is examined in Table 14. The optimal value $\beta$=0.30 effectively balances these competing objectives, while lower values (0.10) provide insufficient regularization and higher values (0.50–0.70) over-constrain the incremental catalyst. As shown in Table 15, preserving the top-$|\mathcal{K}|$=20 strongest off-diagonal correlations achieves the best balance between maintaining important feature relationships and avoiding excessive orthogonalization. Table 16 demonstrates that the base penalty $\lambda_{\text{rest}}$=$5\times10^{-3}$ optimally suppresses cross-dimensional correlations. Weaker penalties fail to adequately regularize the feature space, while stronger penalties excessively suppress meaningful correlations. The penalty ratio analysis in Table 17 confirms that moderately down-weighting the strongest correlations ($\lambda_{\text{topk}}/\lambda_{\text{rest}}$=0.20) provides consistent benefits. Both smaller (0.10) and larger ratios (0.50) yield suboptimal performance. Finally, Table 18 shows that logit-adjusted calibration achieves optimal performance at $\tau = 1.0$. This indicates that moderate prior-shift effectively addresses the long-tailed class distribution, while more extreme temperature settings degrade calibration effectiveness.

## D Hybrid-DomainNet Construction

The Hybrid-DomainNet benchmark is derived from the public DomainNet dataset and specifically designed to evaluate cross-domain generalization under long-tail class-incremental learning con-

Table 14: Sensitivity analysis of $\beta$ (CoD weight).

| $\beta$ | Clipart+Info. | Info.+Quick. |
|---|---|---|
| 0.10 | 59.78 | 45.31 |
| **0.30** | **60.76** | **46.27** |
| 0.50 | 60.19 | 46.01 |
| 0.70 | 59.23 | 45.42 |

Table 15: Sensitivity analysis of $|\mathcal{K}|$ (top-$K$ off-diagonals) in CoD.

| $|\mathcal{K}|$ | Clipart+Info. | Info.+Quick. |
|---|---|---|
| 10 | 60.51 | 46.12 |
| **20** | **60.76** | **46.27** |
| 50 | 60.39 | 46.01 |
| 100 | 60.01 | 45.73 |

Table 16: Sensitivity analysis of $\lambda_{\text{rest}}$ (CoD base penalty).

| $\lambda_{\text{rest}}$ | $1\times10^{-4}$ | $5\times10^{-4}$ | $1\times10^{-3}$ | $5\times10^{-3}$ | $1\times10^{-2}$ |
|---|---|---|---|---|---|
| Clipart+Info. | 59.97 | 60.31 | 60.59 | **60.76** | 60.11 |
| Info.+Quick. | 45.66 | 45.92 | 46.08 | **46.27** | 45.78 |

Table 17: Sensitivity analysis of $\lambda_{\text{topk}}/\lambda_{\text{rest}}$ penalty ratio.

| $\lambda_{\text{topk}}/\lambda_{\text{rest}}$ | Clipart+Info. | Info.+Quick. |
|---|---|---|
| 0.10 | 60.61 | 46.18 |
| **0.20** | **60.76** | **46.27** |
| 0.30 | 60.67 | 46.20 |
| 0.50 | 60.21 | 45.89 |

Table 18: Sensitivity analysis of calibration temperature $\tau$.

| $\tau$ | Clipart+Info. | Info.+Quick. |
|---|---|---|
| 0.5 | 60.32 | 46.02 |
| **1.0** | **60.76** | **46.27** |
| 1.5 | 60.19 | 46.12 |
| 2.0 | 59.64 | 45.58 |

straints. Our construction adheres to three primary objectives: (i) inducing substantial cross-domain shift while maintaining label alignment across domains; (ii) preserving a controllable, ordered long-tailed distribution within each task; and (iii) enabling reproducible evaluation across both 5-task and 10-task incremental learning schedules.

### D.1 DOMAIN SELECTION AND PARTITION STRATEGY

We select three heterogeneous domains that exhibit distinct visual characteristics: Clipart (stylized graphics), Infograph (vector diagrams with text and iconography), and Quickdraw (abstract sketches). These domains provide significant variations in low-level statistics and semantic rendering styles while ensuring stable label alignment without confounding factors such as heavy photo-realism.

The benchmark comprises four distinct partitions to cover both pairwise and multi-domain mixtures:

- Clipart+Infograph: Combines stylized graphics with diagrammatic shapes, exhibiting moderate-to-strong domain shift.

- Clipart+Quickdraw: Pairs stylized color renderings with abstract line drawings, representing strong domain shift.

- Infograph+Quickdraw: Contrasts symbol-rich diagrams with geometric sketches, emphasizing structural differences.

- Clipart+Infograph+Quickdraw: Three-domain mixture providing the most diverse and challenging setting.

This partition strategy balances comprehensive coverage with practical evaluation feasibility, offering a graded spectrum of cross-domain difficulty levels.

### D.2 CLASS SELECTION AND TASK ORGANIZATION

Class subsets are carefully curated from the original DomainNet label space based on four criteria: (i) consistent presence across selected domains within each partition; (ii) taxonomic consistency and annotation reliability; (iii) sufficient sample counts to support imbalanced learning without extreme tail distributions; and (iv) low semantic ambiguity under cross-domain rendering conditions. This selection process yields discriminative class subsets with high cross-domain label fidelity.

For each partition, classes are divided into a base task containing 50% of classes, followed by a sequence of incremental tasks (4 tasks for 5-task schedules, 9 tasks for 10-task schedules). Labels remain strictly disjoint across tasks, and the domain set remains fixed throughout the incremental learning process. All experiments are conducted under exemplar-free constraints.

### D.3 LONG-TAILED DISTRIBUTION PROTOCOL

Within each task, classes follow an ordered long-tailed distribution controlled by an imbalance ratio $\rho \in (0, 1]$:

$$n_c = n_{\max} \, \rho^{\frac{\mathrm{rank}(c) - 1}{|\mathcal{C}_{\text{task}}| - 1}},$$

where $n_{\max}$ represents the head-class sample count, $\mathrm{rank}(c)$ denotes the class rank in the long-tailed ordering, and $|\mathcal{C}_{\text{task}}|$ is the number of classes in the task. For multi-domain training, each class sample count $n_c$ is distributed across domains using fixed mixture proportions that remain constant across tasks, preventing confounding effects between domain shift and changing domain priors. Sampling is performed without replacement while respecting per-domain availability constraints.

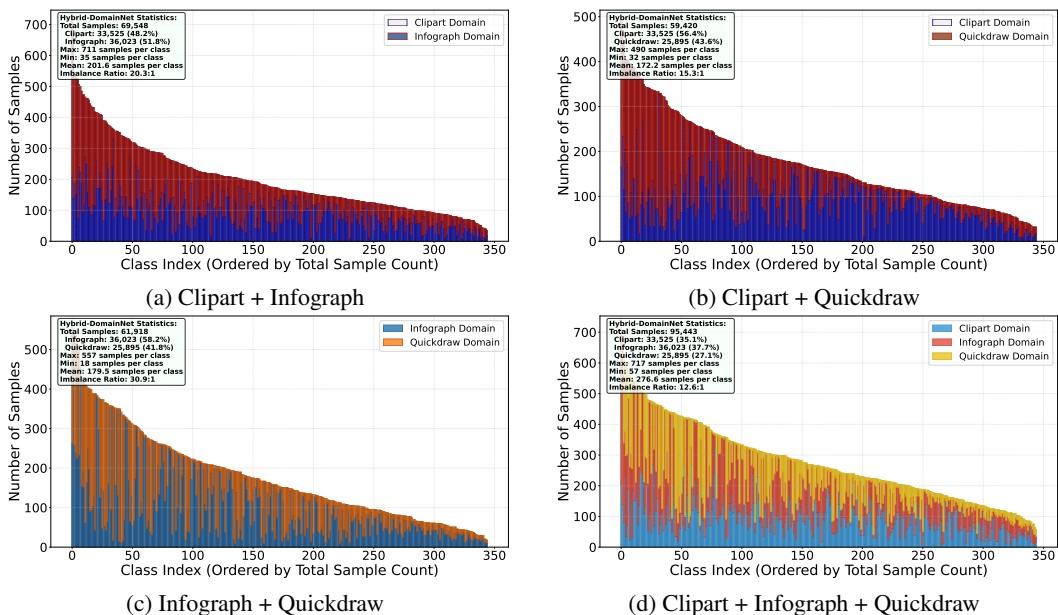

(a) Clipart + Infograph      (b) Clipart + Quickdraw

(c) Infograph + Quickdraw      (d) Clipart + Infograph + Quickdraw

Figure 6: **Class distribution visualizations for Hybrid-DomainNet partitions.** Each panel shows the ordered long-tailed distribution with domain contributions color-coded. Zoomed for better view.

### D.4 DATA PROCESSING AND EVALUATION PROTOCOL

All images undergo consistent preprocessing, including resizing and normalization to match the input requirements of the backbone network. We apply standard data augmentation techniques (random cropping, resizing, and horizontal flipping) uniformly across all domains to prevent domain-specific augmentation biases.

The evaluation protocol employs a balanced cross-domain test set constructed by per-class, per-domain balanced sampling after each task. This approach eliminates evaluation-time biases from class or domain priors. Performance is reported as average accuracy across all tasks. For calibration analyses, class-frequency priors are computed from the current training set and applied exclusively to the corresponding task's logit adjustments.

### D.5 RATIONALE FOR PARTITION DESIGN

The four mixture configurations serve distinct diagnostic purposes: the pairwise mixtures isolate specific domain discrepancy types (stylization vs. iconography, stylization vs. abstraction, iconog-

raphy vs. abstraction), enabling targeted analysis of representation transfer capabilities. The three-domain mixture tests scalability and interference handling capabilities under maximum heterogeneity. Collectively, these settings provide comprehensive coverage of cross-domain learning challenges while maintaining benchmark tractability and reproducibility.

Figure 6 visualizes the class distributions across the four Hybrid-DomainNet partitions under the ordered long-tailed protocol. Bars are sorted by total per-class sample counts, with colors indicating domain contributions. Figure 6a (Clipart+Infograph) demonstrates complementary domain composition with moderate domain gap. Figure 6b (Clipart+Quickdraw) exhibits stronger domain shift between stylized and abstract representations. Figure 6c (Infograph+Quickdraw) presents the most challenging pairwise mixture due to distinct low-level statistics. Figure 6d (three-domain mixture) showcases the richest cross-domain variability and diverse per-class domain composition.

## E  LIMITATIONS

Our proposed approach, while effective in the evaluated scenarios, inherits certain limitations from its underlying technical foundations. As a method built upon pre-trained models (PTMs) using parameter-efficient tuning (PET) techniques, our framework requires a powerful feature extractor to fully leverage the generalization capabilities of PTMs. This dependency may limit applicability in scenarios requiring training from scratch or when dealing with very small-scale tasks where PTM advantages are less pronounced.

Although our experimental results demonstrate that a single set of default hyperparameters for Bi-level Contrastive Learning (BCL) and Correlation Distillation (CoD) performs robustly across both LTCIL and CD-LTCIL settings, these default configurations may not achieve optimal performance when applied to datasets with substantially different statistical characteristics. For instance, adaptation to specialized domains such as medical imaging may require targeted hyperparameter optimization.

More importantly, our current framework addresses cross-domain incremental learning under domain-consistent conditions, where the domain set remains invariant throughout all tasks. A more challenging and realistic scenario involves dynamic domain shifts across tasks, where entirely new domains emerge during the incremental learning process. Addressing such dynamic domain-shift scenarios presents significant challenges for LTCIL and represents an important direction for future research.

## F  DECLARATION OF LARGE LANGUAGE MODEL USAGE

We transparently disclose that large language models (LLMs) were utilized exclusively as writing assistants to enhance the clarity and polish of this manuscript. The following statements specify the precise scope of LLM involvement and the safeguards implemented to ensure research integrity.

**LLMs were employed solely for language refinement purposes,** including: (i) improving grammatical accuracy, writing style, and overall readability of author-composed text; (ii) harmonizing terminology usage and enhancing section transitions; (iii) refining figure and table captions and section headings; and (iv) condensing or eliminating redundant prose while rigorously preserving original technical meaning.

**Crucially, LLMs were not involved in any core research activities,** including research ideation, problem formulation, dataset construction, model or algorithm design, mathematical derivation, hyperparameter selection, code implementation, experimental execution, result computation, or scientific conclusion drawing. All technical content—including definitions, equations, loss functions, training schedules, and reported numerical results—originates entirely from the authors' original research, implementations, and experiments.

The usage process involved strictly constrained prompts focused exclusively on paraphrasing and copy-editing of author-provided text segments. All LLM-generated suggestions underwent thorough review, selective editing, or complete rejection by the authors. Technical statements, mathematical symbols, and equations were systematically cross-verified against source materials and implementations. No non-public data, personal information, or proprietary code was disclosed to LLM services.

**Research reproducibility and integrity were maintained** throughout: all experimental results, performance metrics, and visualizations were generated exclusively by the authors' code and computational pipelines. LLMs exerted no influence on experimental outcomes or methodological choices, with complete configuration details documented in the Appendix. The scientific content and claims presented in this work remain entirely unaffected by LLM usage.

Regarding specific tools, generic off-the-shelf LLM services were employed strictly for the language polishing purposes described above. No specialized model outputs constitute part of the scientific contribution of this paper.

