# OpenReview forum: "Contrastive-and-Correlation Catalysts for Cross-Domain Long-Tail Class-incremental Learning"
_ICLR.cc/2026/Conference — ICLR 2026 Conference Withdrawn Submission_

### Official Review · Reviewer_Ssh4 · 2025-10-14

**Soundness:** 2
**Presentation:** 3
**Contribution:** 2
**Rating:** 4
**Confidence:** 4

**Summary:**

This paper proposes a cross-domain long-tail class-incremental learning problem and proposes its solution, namely C2C using the idea of catalysts and bi-level contrastive learning.

**Strengths:**

1) the problem of cross-domain long-tailed class-incremental learning appears to be new;
2) the proposed C2C method performs well;

**Weaknesses:**

1) cross-domain continual learning refers to a specific setting in continual learning. please see references below. you need to rename your problem.

2) this statement is incorrect "both conventional CIL and LTCIL remain constrained by an unrealistic assumption: all incremental task share the same domain". The problem of cross-domain continual learning has been explored before this work [1], [2]. please compare how your problem is different from these prior works.

3) literature survey is outdated. You should also review recent advances in parameter-efficient fine-tuning approaches for continual learning especially those published in 2024 onward such as [3].

4) I have a doubt how the domain shift occurs in the evaluation phase. if a model is given a task coming from multiple domains with their labels, in such as case, not even domain alignment needs to be done. How do you evaluate your model? are these testing samples drawn from the same domains as the training case?

5) I also have a doubt regarding to practical setting of this problem. How does this setting reflect real-world scenario?

6) how does the catalyst differ from LoRA or Adapter? if they don't significantly differ from them, there is no urgency to give a new name.

7) The domain alignment via contrastive learning is done only in task 1. Is there any guarantees that these aligned domain remain robust given sequential tasks.

8) The experimental setup is not sufficiently challenging because you reserve half of the classes for the base task. I wonder if this base task size shrinks.


[1] M. De Carvalho, M. Pratama, J. Zhang, H. Chua, E. Y. Kien Yee, Towards Cross-Domain Continual Learning, 40th IEEE International Conference on Data Engineering (ICDE 2024)
[2] W. Weng, M. Pratama, J. Zhang, C. Chen, E. Yapp Kien-Yee, R. Savitha, Cross-Domain Continual Learning via CLAMP, Information Sciences, 2024
[3] M. A. Masum, M. Pratama, R. Savitha, L. Liu, Habibullah, R. Kowalczyk, Vision and Language Synergy for Rehearsal Free Continual Learning, The Thirteenth International Conference on Learning Representations (ICLR 2025), 2025

**Questions:**

1) cross-domain continual learning refers to a specific setting in continual learning. please see references below. you need to rename your problem.

2) this statement is incorrect "both conventional CIL and LTCIL remain constrained by an unrealistic assumption: all incremental task share the same domain". The problem of cross-domain continual learning has been explored before this work [1], [2]. please compare how your problem is different from these prior works.

3) literature survey is outdated. You should also review recent advances in parameter-efficient fine-tuning approaches for continual learning especially those published in 2024 onward such as [3].

4) I have a doubt how the domain shift occurs in the evaluation phase. if a model is given a task coming from multiple domains with their labels, in such as case, not even domain alignment needs to be done. How do you evaluate your model? are these testing samples drawn from the same domains as the training case?

5) I also have a doubt regarding to practical setting of this problem. How does this setting reflect real-world scenario?

6) how does the catalyst differ from LoRA or Adapter? if they don't significantly differ from them, there is no urgency to give a new name.

7) The domain alignment via contrastive learning is done only in task 1. Is there any guarantees that these aligned domain remain robust given sequential tasks.

8) The experimental setup is not sufficiently challenging because you reserve half of the classes for the base task. I wonder if this base task size shrinks.


[1] M. De Carvalho, M. Pratama, J. Zhang, H. Chua, E. Y. Kien Yee, Towards Cross-Domain Continual Learning, 40th IEEE International Conference on Data Engineering (ICDE 2024)
[2] W. Weng, M. Pratama, J. Zhang, C. Chen, E. Yapp Kien-Yee, R. Savitha, Cross-Domain Continual Learning via CLAMP, Information Sciences, 2024
[3] M. A. Masum, M. Pratama, R. Savitha, L. Liu, Habibullah, R. Kowalczyk, Vision and Language Synergy for Rehearsal Free Continual Learning, The Thirteenth International Conference on Learning Representations (ICLR 2025), 2025

---

### Official Review · Reviewer_Akr7 · 2025-10-25

**Soundness:** 2
**Presentation:** 3
**Contribution:** 2
**Rating:** 2
**Confidence:** 4

**Summary:**

The paper introduces a new setting called Cross-Domain Long-Tail Class-Incremental Learning (CD-LTCIL) and proposes a combined framework, C2C (Contrastive-and-Correlation Catalysts). The method integrates contrastive transfer, correlation-based distillation, and logit calibration under a frozen pretrained backbone. Experiments on CIFAR-100, ImageNet-R, and a newly constructed “Hybrid-DomainNet” dataset show strong results.

**Strengths:**

1. The proposed setting (CD-LTCIL) is novel at least as a formal definition, extending long-tail CIL to cross-domain scenarios.
2. The method achieves competitive or superior performance on several benchmarks.
3. Implementation and experimental sections are well organized and reproducible.

**Weaknesses:**

1. The problem formulation feels artificial and not well motivated by a real-world use case. The cross-domain + long-tail combination seems synthetically constructed rather than naturally arising.
2. The technical design of C2C heavily borrows existing mechanisms (contrastive alignment, correlation distillation, logit adjustment, adapter tuning). The integration lacks clear theoretical or intuitive justification.
3. The paper gives the impression of technique aggregation rather than conceptual innovation. None of the components are explained with new insights or backed by analysis beyond heuristics.
4. Citations are often missing where prior ideas are reused, creating ambiguity about what is genuinely novel.
5. The contribution boundary between the new setting, method, and dataset is unclear—each is somewhat incremental.
6. It seems that there is a branch of study on "cross-domain continual learning [1]", at which this paper takes its position. However, the paper does not include any discussion on the relatedness of this work to existing cross-domain continual learning methods.

[1] De Carvalho, Marcus, et al. "Towards cross-domain continual learning." 2024 IEEE 40th International Conference on Data Engineering (ICDE). IEEE, 2024.

**Questions:**

1. Please provide stronger motivation or practical scenario where CD-LTCIL naturally arises.
2. What makes the proposed combination non-trivial beyond stacking existing modules?
3. Please add proper attribution for reused ideas (contrastive transfer, correlation distillation, logit adjustment) from existing works if any.
4. Why didn't you first study CD-CIL, as CD-CIL seems to be a more fundamental version than CD-LTCIL?

---

### Official Review · Reviewer_KfKt · 2025-10-29

**Soundness:** 3
**Presentation:** 3
**Contribution:** 2
**Rating:** 4
**Confidence:** 4

**Summary:**

The authors propose a new Cross-Domain Long-Tail Class-Incremental Learning (CD-LTCIL) setting. In this setting, each incremental task contains data from multiple heterogeneous domains with a long-tailed distribution, and the model must simultaneously tackle catastrophic forgetting, cross-domain shift, and class imbalance. To solve this problem, the authors design the parameter-efficient C2C (Contrastive-and-Correlation Catalysts) framework: it freezes a pre-trained backbone to retain stable features and uses lightweight "catalyst" modules for domain adaptation. Experiments show that C2C significantly outperforms state-of-the-art methods such as APART in both CD-LTCIL (on Hybrid-DomainNet) and standard LTCIL (on CIFAR-100, ImageNet-R) scenarios, and ablation studies validate the effectiveness of its core components.

**Strengths:**

- The paper proposes the Cross-Domain Long-Tail Class-Incremental Learning (CD-LTCIL) scenario for the first time, filling the gap of the single-domain assumption in conventional LTCIL.
- The authors adopt the design of "frozen pre-trained backbone + lightweight catalysts", only updating a small number of parameters for cross-domain adaptation, and A+BCL addresses cross-domain shift, CoD mitigates catastrophic forgetting, and LTC solves class imbalance.
- The authors construct the Hybrid-DomainNet dedicated benchmark, covering multi-domain combinations and long-tailed distributions; compare with 8 SOTA methods across 3 categories, verifying generalization in both CD-LTCIL and standard LTCIL scenarios.

**Weaknesses:**

- C2C relies on high-quality pre-trained backbone networks (only ViT-B/16 is validated) and fails to explore performance in scenarios with weak pre-training or no pre-training. Additionally, the CD-LTCIL setting assumes a fixed set of domains, making it unable to address the challenge of "dynamic domain shift" (incremental tasks involving entirely new domains) in real-world scenarios.
- The intensity of cross-domain shift is not quantified (e.g., no use of metrics like MMD or WD), making it impossible to accurately evaluate C2C’s adaptability to shifts of varying intensities. Only "ordered long-tailed distributions" are validated, with no testing of robustness under random or extreme long-tailed distributions.
- Hybrid-DomainNet is only derived from DomainNet (focused on daily object categories) and does not cover specialized domains such as medical imaging or remote sensing. In the discussion section, there is no in-depth comparison of core differences between C2C and existing methods, and the guidance for future research directions is limited.

**Questions:**

- The paper only validated with ViT-B/16 (pre-trained on ImageNet21K), lacking performance experiments for weak pre-training (e.g., ViT pre-trained on CIFAR-100) and non-pre-training scenarios. What if C2C with a lightweight backbone?
- CD-LTCIL assumes fixed domains, lacking performance degradation data for dynamic domain shift (new domains in incremental tasks) and preliminary experiments on adaptation schemes like "dynamic base catalyst update". Can you conduct experiments on domain incremental?
- As the core adaptation component, lacking ablation experiments for catalyst variants (different bottleneck dimensions, insertion positions) and experimental support for the basis of current structure selection.
-  Lacking quantitative experiments on cross-domain shift intensity for each Hybrid-DomainNet partition and comparison experiments on C2C’s adaptability to shifts of varying intensities.
-  lacking performance experiments for random/extreme long-tailed (imbalance ratio 100:1) scenarios and comparison experiments between LTC and long-tail methods like re-sampling/re-weighting.

---

### Official Review · Reviewer_iotY · 2025-10-30

**Soundness:** 3
**Presentation:** 3
**Contribution:** 3
**Rating:** 4
**Confidence:** 3

**Summary:**

This paper introduces a novel continual learning setting called Cross-Domain Long-Tail Class-Incremental Learning (CD-LTCIL). CD-LTCIL makes the models sequentially learn new classes from long-tailed distributions, with each incremental task comprising data from multiple heterogeneous domains.
Then the paper proposes C2C (Contrastive-and-Correlation Catalysts), which utilizes a frozen pre-trained backbone augmented with lightweight "catalyst" pathways. For the initial task, it employs cosine anchoring and bi-level contrastive learning to learn domain-invariant class representations. For subsequent tasks, it preserves knowledge through cross-correlation distillation between a frozen base catalyst and a learnable incremental one, further incorporating logit-adjusted calibration to handle class imbalance. Extensive experiments on both standard LTCIL benchmarks and the newly introduced Hybrid-DomainNet demonstrate that C2C achieves state-of-the-art performance.

**Strengths:**

This paper introduces a novel and more realistic learning paradigm that challenges models to continually learn new classes from long-tailed and heterogeneous domains while mitigating catastrophic forgetting.

To provide a robust foundation for evaluating and advancing cross-domain long-tail class-incremental learning, this work constructs Hybrid-DomainNet, a new benchmark specifically designed for this intricate setting.

C2C is a parameter-efficient and exemplar-free framework that effectively tackles domain shifts, class increments, and long-tail distributions simultaneously through contrastive alignment and correlation distillation, without requiring previous task data.

**Weaknesses:**

- Issues with Writing and Contribution Clarity: The introduction primarily highlights the shortcomings of existing methods (e.g., LTCIL's single-domain assumption) and the need for the proposed Cross-Domain Long-Tail Class-Incremental Learning (CD-LTCIL) setting. However, it fails to clearly articulate the specific technical contributions and novelties of their proposed C2C framework beyond a high-level overview. The explanation of how C2C uniquely addresses these identified challenges and what specific mechanisms are introduced to achieve this remains vague. Similarly, the "Related Work" section provides broad categorizations of existing methods but lacks a deep analysis of their technical underpinnings or a clear exposition of how C2C fundamentally differentiates itself from these prior approaches. This makes it difficult for a reader to immediately grasp the paper's original technical advancements.

- The paper states that for the initial task (Task 1), Contrastive Transfer Learning (CA) and Bi-level Contrastive Learning (BCL) are employed. What are the precise objectives of each of these losses? Is the primary goal better domain adaptation, or is it to more effectively preserve knowledge from the pre-trained model, or both? Crucially, why are these specific loss functions (CA + BCL) exclusively applied during Task 1? If they are effective in learning "domain-invariant class representations" and "domain-robust representations", why are they not utilized or adapted for subsequent incremental tasks where domain shifts and long-tail distributions persist?

- What are the fundamental differences between this correlation-based distillation and more conventional feature-based distillation techniques (e.g., L2 loss on feature maps, attention map distillation)? What is the underlying motivation for designing distillation based on correlation matrices rather than direct feature comparison? Why is preserving "higher-order statistical relationships" (L257) hypothesized to be more effective than other forms of knowledge transfer in this specific context?

- Have the authors conducted ablation studies or comparisons against established distillation baselines (e.g., knowledge distillation, feature distillation) to empirically validate the superiority or unique benefits of CoD? Without such comparisons, it is hard to ascertain if CoD offers a genuine technical innovation or if it might be an over-engineered solution.

- The paper states regarding the correlation distillation loss (L268-269): "The first term encourages one-to-one alignment between corresponding feature dimensions to maintain stability. The second term penalizes off-diagonal correlations to prevent representational collapse and preserve plasticity for learning new concepts." What is the specific mechanism by which penalizing "off-diagonal correlations" prevents "representational collapse" and simultaneously "preserves plasticity"? Are there any prior theoretical works or established principles in deep learning that provide a strong foundation or empirical evidence for these claimed effects of correlation matrix manipulation?

- In the "Analysis of Correlation Distillation" (L433-438) and Tables 14-17, the paper investigates the effect of CoD components. Are these observed characteristics of correlation distillation (e.g., its ability to balance stability and plasticity, or to prevent representational collapse) a general property inherent to how PTMs and PET methods learn and adapt in incremental settings, or are they specific to the C2C framework?Do similar correlation-based analyses or regularization techniques exist and show comparable effects in other PTM/PET-based continual learning methods? Understanding this could clarify whether CoD's insights are a novel C2C-specific mechanism or a broader characteristic of PTM adaptation.

- While the proposed Cross-Domain Long-Tail Class-Incremental Learning (CD-LTCIL) setting undeniably increases the difficulty for incremental learning models, its novelty could be questioned. The concept of integrating multiple domains into continual learning, or dealing with long-tailed distributions in CIL, are not entirely new independently.

- Is the combination of cross-domain, long-tail, and class-incremental learning truly a novel setting, or is it primarily a re-packaging/re-arrangement of existing challenges? The paper states the Hybrid-DomainNet benchmark is "derived from the public DomainNet dataset" (L317, L808). Is the construction process primarily a re-sorting and partitioning of an existing dataset to fit this combined challenge, rather than introducing entirely new data characteristics or a fundamentally new problem formulation?


- Despite the ablation studies suggesting some insensitivity to certain hyperparameters, the sheer volume of parameters raises concerns about the complexity of tuning and the robustness of the method across diverse scenarios not covered in the experiments.

- It is crucial to clearly define the specific innovations within C2C's architecture and learning objectives. Without this explicit clarification, it's difficult to assess the true technical novelty and intellectual contribution of the proposed method.


- How can the reproducibility and fairness of these reimplementations be guaranteed without publicly available code or detailed implementation specifics for all compared baselines? Is there an established, independent benchmark or shared code repository that provides a standardized implementation of these SOTA methods under the CD-LTCIL setting?

**Questions:**

See above.

---

### Note · Authors · 2025-11-12

I have read and agree with the venue's withdrawal policy on behalf of myself and my co-authors.